# Recent Advancements in Enhancing Antimicrobial Activity of Plant-Derived Polyphenols by Biochemical Means

**Likun Panda [1,2] and Arturo Duarte-Sierra [1,2,3,*]** 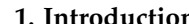

1 Food Science Department, Laval University, Quebec, QC G1V 0A6, Canada; likun.panda.1@ulaval.ca
2 Institute on Nutrition and Functional Foods (INAF), Laval University, Quebec, QC G1V 0A6, Canada
3 Center for Research in Plant Innovation (CRIV), Laval University, Quebec, QC G1V 0A6, Canada
* Correspondence: arturo.duarte-sierra@fsaa.ulaval.ca

**Abstract:** Plants are a reservoir of phytochemicals, which are known to possess several beneficial health properties. Along with all the secondary metabolites, polyphenols have emerged as potential replacements for synthetic additives due to their lower toxicity and fewer side effects. However, controlling microbial growth using these preservatives requires very high doses of plant-derived compounds, which limits their use to only specific conditions. Their use at high concentrations leads to unavoidable changes in the organoleptic properties of foods. Therefore, the biochemical modification of natural preservatives can be a promising alternative to enhance the antimicrobial efficacy of plant-derived compounds/polyphenols. Amongst these modifications, low concentration of ascorbic acid (AA)–Cu (II), degradation products of ascorbic acid (DPAA), Maillard reaction products (MRPs), laccase–mediator (Lac–Med) and horse radish peroxidase (HRP)–$H_2O_2$ systems standout. This review reveals the importance of plant polyphenols, their role as antimicrobial agents, the mechanism of the biochemical methods and the ways these methods may be used in enhancing the antimicrobial potency of the plant polyphenols. Ultimately, this study may act as a base for the development of potent antimicrobial agents that may find their use in food applications.

**Keywords:** polyphenols; antimicrobial activity enhancement; ascorbic acid; Maillard reaction products; laccase–mediator system; horseradish peroxidase–$H_2O_2$ system

## 1. Introduction

The quality and safety of food products are compromised by the loss of nutrients, sensory attributes, and microbial growth. The prior can be in terms of off odor, off flavor, discoloration, or production of toxic compounds as metabolic end products of the saccharolytic, proteolytic, pectinolytic, and lipolytic enzymes, which ultimately leads to food poisoning or intoxication. Ensuring food safety and meeting the demand for food without synthetic chemical preservatives has led to increased interest in natural alternatives to inactivate microorganisms and enzymes in food [1,2]. Although attempts have been made to produce additive-free foods, it is unlikely that the current marketing system could exist without the use of antimicrobials. In addition, requirements for toxicology safety have limited the ability of the industry to develop new chemical antimicrobials [3]. Therefore, it is essential for the food industry to find new and natural antimicrobial food alternatives.

Phytochemicals can be recovered from plant products and used as ingredients in food and cosmetics, as healthy antimicrobials, and as alternatives to chemical preservatives. A typical feature of plants is their ability to synthesize a wide range of phyto-compounds (i.e., secondary metabolites), which play essential roles in the interaction of the plant with its environment [4]. They can be structurally divided into five major groups: phenylpropanoids, flavonoids, polyketides, terpenoids, and alkaloids. Furthermore, it is increasingly clear that several phytocompounds in fruits and vegetables of different chemical classes are beneficial to human health [5,6]. Interestingly, several phytochemicals such as simple phenolic acids,

polyphenols, terpenes, isothiocyanates, polyacetylenes, etc., also exhibit antimicrobial properties. There is sufficient evidence supporting the potential of plant-derived phytochemicals as natural antimicrobial agents [7–9].

However, only a few natural antimicrobials have found practical application in the food industry. Their use in foods as preservatives is often limited due to the need for high concentrations to achieve the desired activity, which may modify the sensory characteristics of food by making it unacceptable [10]. Another limitation is the interaction of natural antimicrobial with complex food matrices, mainly with hydrophobic compounds such as lipids [11]. Nonetheless, polyphenols can interact with proteins through hydrophobic or hydrophilic interactions, leading to the formation of soluble or insoluble complexes [12].

The "antimicrobial potency" of polyphenols can be altered and enhanced by biochemical means that could allow their application as antimicrobial agents. This means that the antimicrobial effect can be improved by reducing the effective concentration of the plant-derived compounds [13]. Biochemical and physiological studies have provided a large body of evidence to surmise that plant-derived polyphenols can be well adapted to achieve promising and potent antimicrobials for use in foods, ensuring microbial safety of foods without chemical additives [14,15]. Such biochemically modified natural ingredients would positively affect food preservation without compromising the sensory attributes and health of the consumers.

## 2. Plant Polyphenols as Antimicrobials

Recent studies have shown that plant compounds used as natural antimicrobials are safe alternatives to chemical additives [16,17]. Natural antimicrobials' mechanism of action includes cell membrane rupture, defective nucleic acid mechanisms, decay of the proton motive force, and depletion of adenosine triphosphate (ATP). The antimicrobials from plants (polyphenols, essential oils) use the aforementioned mechanisms of action against foodborne bacteria [18]. Amongst all secondary metabolites in plants, polyphenols are the ones that play multiple essential roles in plant physiology, also in addition having potential health-benefiting properties such as having antioxidant, antiallergic, anti-inflammatory, anticancer, antihypertensive, and antimicrobial features [19,20]. Basically, they are divided into flavonoids and non-flavonoids, on the basis of their chemical structure.

### 2.1. Flavonoids

Flavonoids, such as catechins, flavones, and flavonols, have antifungal, antiviral, and antibacterial activities [21]. The antimicrobial activity of the flavonoid quercetin is attributed to the inhibition of the enzyme DNA gyrase and (-)-epigallocatechin gallate, which was reported to inhibit the energy metabolism [21]. Flavonoids, especially catechins and proanthocyanidins (due to antioxidant properties), have been proposed to neutralize bacterial toxic factors originating from *Vibrio cholerae, Staphylococcus aureus, Vibrio vulnificus, Bacillus anthracis, Clostridium botulinum* [22]. The citrus flavonoids, such as apigenin, kaempferol, quercetin, and naringenin, are effective antagonists of cell–cell signaling [23]. Recent reviews have provided lines of evidences on the antimicrobial activity of plant flavonoids along with their mechanism of actions [24,25].

### 2.2. Non-Flavonoids

Phenolic acids (benzoic, phenylacetic, and phenylpropionic acids) have been found to inhibit pathogenic and non-pathogenic bacteria and fungi such as *Escherichia coli, Lactobacillus* spp., *S. aureus, Pseudomonas aeruginosa* and *Candida albicans* [26]. Hydroxycinnamic acids (caffeic, coumaric, ferulic, and sinapic acids) have been found to inhibit *Bacillus cereus* and *S. aureus*; *P. fluorescens* [27]. In addition, the antibacterial activity of caffeic, ferulic, and p-coumaric acids against *E. coli*, *S. aureus*, and *B. cereus*, with p-coumaric acid being effective against *E. coli*, has been reported [27]. Hydroxycinnamic acids (i.e., nitrobenzoate, p-aminobenzoate, ethyl aminobenzoate, ethyl- and methyl-benzoate, salicylic acid, transcinnamic acid, trans-cinnamaldehyde, ferulic acid, o-acetoxy benzoic acid, and anthranilic

acid) have been found to inhibit aflatoxins production from *Aspergillus flavus* and *Aspergillus parasiticus* [28]. Additionally, furocoumarins present in carrots, celery, citrus fruits, parsley, and parsnips have been reported for their antimicrobial activity against *E. coli* O157:H7, *Erwinia carotovora*, *Listeria monocytogenes*, and *Micrococcus luteus* [29].

The antibacterial properties of some common foods and beverages such as coffee against *Legionella pneumophila* and *E. coli* O157:H7 are attributed to its compounds such as caffeic acid, chlorogenic acid, and protocatechuic acid [30,31]. Furthermore, tea (*Camellia sinensis*) has also been found to display antimicrobial properties [32–34] through its predominant catechin, epigallocatechin gallate, against methicillin-resistant *S. aureus* (MRSA). The compound E-cinnamaldehyde has been found to significantly contribute to the antimicrobial properties of cinnamon stick extract (Ext) against *B. cereus, E. coli, L. monocytogenes, S. aureus,* and *Salmonella* [35].

### 2.3. Extraction of Polyphenols from Plant Products

Extraction methods have been developed recently using modern technology. These methods use fewer or no organic solvents, thereby minimizing environmental and health impacts and maximizing the yield of desired polyphenols by selective extraction [36]. Advanced methods such as microwave-assisted, ultrasound-assisted, pulsed-electric-field-assisted and enzyme-assisted extractions, as well as pressurized liquid and supercritical fluid extractions, are given prime importance these days to extract desired polyphenols from the plant products [37,38]. One of the recent studies has suggested extraction of non-extractable or bound polyphenols by pretreatment using the aforementioned methods, which are further cleaved using acid, alkaline, or enzyme treatments, followed by purification step using solid-phase extraction column chromatography and finally storage step using lyophilization [39]. Studies have illustrated that the bioavailability and yield of polyphenols are one of the most important factors of their antimicrobial activity [40,41]. However, along with these factors, their structure has also been found to play a critical role in their antimicrobial activity [42,43]. The relationship between the structure of polyphenols and their antimicrobial activity is elaborately illustrated in the proceeding section.

### 3. Antimicrobial Activity and Structural Relationship of Plant-Derived Polyphenols

The structural diversity of polyphenols is immense, and the impact of antimicrobial action they produce against microorganisms depends on their structural configuration [44]. For instance, Phenolic acids inhibit the activity of bacterial enzymes, disrupting their metabolism and depriving the substrates necessary for growth. The hydroxycinnamic acids (p-coumaric acid, caffeic, and ferulic acid) induced higher ion leakage and a more significant influx of protons into the cells, compared with hydroxybenzoic acids, gallic, vanillic, and syringic acid [45]. Additionally, these hydroxycinnamic acids have been found to meet Lipinski's rules, proving their functional potential as drugs and antimicrobial agents. The relationship between chemical structure and biological activity has received considerable attention in recent years because it allows the prediction of chemical toxicity or bioactivity without an inordinate amount of time and effort.

The potency of an antimicrobial is attributed to its structural characteristics. The relationship of the antimicrobial activity of plant polyphenols is classified into four types: (1) position of functional groups (FNG), (2) number of FNG, (3) presence of C2=C3 double bond, and (4) type of FNG.

### 3.1. Position of Functional Group

The structural antimicrobial activity of the major plant polyphenols, i.e., flavonoids, is well documented [46]. The amphipathic features of flavonoids play an essential role as far as antibacterial properties are concerned [47]. The hydrophobic substituents such as prenyl groups, alkylamino chains, alkyl chains, and nitrogen or oxygen-containing heterocyclic moieties usually enhance the antibacterial activity of all flavonoids [48]. Different classes of

flavonoids, mainly chalcones, flavanes, and flavan-3-ol exhibits better antimicrobial activity due to variation in the position of the functional group attached to the rings [46].

### 3.1.1. Chalcones

Several studies have suggested that chalcones with a lipophilic group such as iso-prenoid and methoxy groups at positions 3′, 5′, and 2′ of ring A are the most potent inhibitors of MRSA strains [49]. Based on the activity of isobavachalcone (MIC: 30 μg/mL), (Figure 1A), the authors of [50] suggested the that A ring with a prenyl group displays adequate antimicrobial activity, but cyclization or addition of the prenyl group to B ring in addition to the A ring decreases this activity. Likewise, the hydroxy group at 4′, 4, and 6 of A and B rings increase the antimicrobial activity. For example, kuraridin and 7,9,2′,4′-tetrahydroxy-8-isopentenyl-5-methoxychalcone (THIPMC) compounds with the same structure, with only one difference in the OH of the B ring (2 and 4 instead of 4 and 6), showed high activity against the MRSA strain [51].

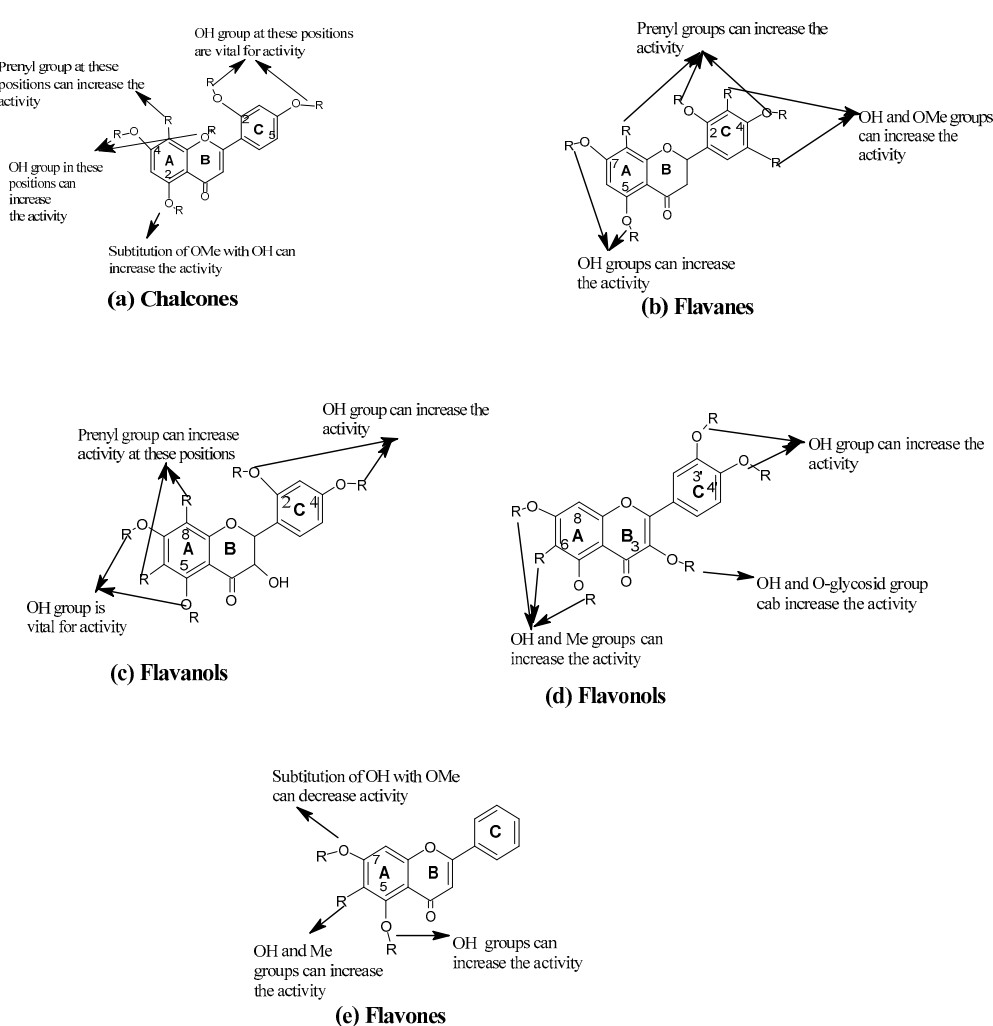

**Figure 1.** Structure–activity (SAR) relationship of important flavonoids: (**a**) in chalcones, substitution of OMe, OH, and prenyl group in ring A, OH group in ring B, and OH groups in 2′ and 4′ position of ring C enhances antimicrobial activity; (**b**) on flavanes, substitution of prenyl and OH groups at 5′ and 7′ positions in ring A and OH, OMe groups at 3′ and 5′ position and prenyl at 4′ position in ring C enhances antimicrobial activity; (**c**) in flavanols, substituting OH group at 5′ position and prenyl at 8′ position of ring A, and OH groups at 2′, 4′ position can enhance antimicrobial activity; (**d**) in flavonols, replacing OH and Me group in ring A, OH, O-glycoside group in ring B and OH groups in ring C can improve antimicrobial activity; (**e**) In flavones substitution of OH group at 5′, OH, Me groups at 6′, and OH, OMe group at 7′ of ring A can improve antimicrobial activity.

### 3.1.2. Flavanes and Flavanols

Flavanes with a prenyl group at the A ring have been found to be the most potent antibacterial compounds against *S. aureus*. It is established that the number and position of prenyl groups on this ring increase antimicrobial activity [52]. The presence of the hydroxy groups at different positions on A and B rings has also been reported to improve antibacterial activity (Figure 1B,C). The compound 3′-O-methydiplacol with OH at the 5, 3′, and 4′ positions of the A and B rings, respectively, as well as the geranyl group at C-6 and OMe at C-5′, showed satisfactory (i.e., MIC value of 4 µg/mL) activity against *S. aureus* [53]. additionally, sophoraflavanone with isogeranyl at C-8 and OH at 3, 2′, and 4′ at A and B rings were active (i.e., MIC value of 7.3 µg/mL) against *S. aureus* [51]. The position of prenyl, hydroxyl, and especially methoxy groups at positions 5 and 7 of the A ring, increased the antibacterial effect of flavanes and flavones [54]. Furthermore, different substitutions on position 3 of the C ring with a hydroxyl or an *O*-glycoside group could enhance the antimicrobial activity of certain flavones [46]. One of the earlier findings also suggested that the tetraflavonoids without OH on the C ring showed moderate activity against *E. coli* [55].

### 3.1.3. Flavonols

In the A ring, many studies have confirmed that hydroxylation at positions 5 and 7 together are critical for the antibacterial activity of flavonols against *S. aureus* strains [56]. In addition, hydroxylation on the B and C rings also increased the antimicrobial activity of these compounds. A comparison of compounds with the same structure showed that kaempferol with a hydroxy group at C-4′ had less activity than galangin (without OH at C-4′) against *S. aureus* (Figure 1D) [47]. The number of glycosyl groups instead of the hydroxy group at position 3 has been found to have a significant effect on antibacterial activity. For example, among the compounds extracted from *Maytenus buchananii*, quercetin-3-O-[α-L-rhamnopyranosyl-(1 → 6)-β-D glucopyranoside] with a disaccharide group at the same position was the better inhibitor of *S. aureus* growth than amentoflavone-7″,4‴-dimethyl-ether with monosaccharide group (quercetin-3-O-β-D-glucopyranoside) [57]. Substitution of the methoxy group at position 3 decreased the antimicrobial activity. For example, piliostigmol (with OMe and Me groups at positions 6 and 7 of the A ring and OH at position 3) was more active against *S. aureus* than 6-C-methylquercetin-3,3′,7-trimethyl ether (with OMe at the C-3 position) [58].

### 3.1.4. Flavones

Studies conducted on the antibacterial activity of flavones [59] suggested that at least one hydroxy group in the A ring (especially at C-7) is vital for antibacterial activity. Hydroxyl groups in other positions such as C-5 and C-6 can also increase the antibacterial action [60]. However, the substitution of OH with OMe at C-7 was seen to reduce the antibacterial activity. For instance, the compound 5,7-dihydroxy-flavone with two OH at positions 5 and 7 has been found to be more potent against *Ralstonia solanacearum* (i.e., MIC: 25 and 300 µg/mL) compared to 5-hydroxy-7-methoxy-flavone with OMe at position 7 and OH at position 5 (Figure 1E) [61]. The importance of the –OH group at position 5 of flavones for their antimicrobial activity against MRSA strains has also been reported [62]. One of the investigations on plant isoflavonones suggested that the hydroxyl group's C-5, 6 and 7 position is crucial for antimicrobial action [63]. The presence of the prenyl (C5) group at position 6 without cyclization of this substituent with the A ring has also been reported to improve antibacterial activity [52].

### 3.2. Number of Functional Groups Attached

The number of functional groups attached has been found to have a significant influence on antimicrobial activity. The 2′, 4′- or 2′, 6′-dihydroxylation of the B ring and 5, 7-dihydroxylation of the A ring in the flavanone structure are essential for anti-MRSA activity [64]. Moreover, 5-hydroxyflavanones and 5-hydroxyisoflavanones with one, two

or three additional hydroxyl groups at the 7′, 2′, and 4′ positions inhibited the growth of *Streptococcus mutans* and *S. sobrinus* [65]. Caffeic acid had higher antimicrobial activity than p-coumaric acid due to the additional –OH group on the phenolic ring of the former compound [66].

### 3.3. Presence of C2=C3 Double Bond

It has been observed that flavanones with C2=C3 are more active than the corresponding flavones. For example, naringenin showed antibacterial effects on all the tested bacteria, whereas apigenin showed almost no effect [67]. In addition, the C2=C3 double bond was found to be responsible for the antifungal activity of 5,7-dihydroxyflavonoids, while hydrogenation of the C2=C3 bond reduced the antifungal effect [68]. The flavonoids apigenin luteolin, dinatin, and daidzein, C2=C3 had better anti-influenza virus activities, compared with catechin and epicatechin belonging to the flavanols class of compounds that lack the C2=C3 bond [69].

### 3.4. Type of Functional Group

The hydrophobic substituents such as prenyl groups, alkylamino chains, alkyl chains, and nitrogen- or oxygen-containing heterocyclic moieties have been reported to enhance the activity of all the flavonoids [48]. Variation in the antimicrobial activity of polyphenols also depends on variation in the functional group they have [70]. The substitution of the phenyl moiety by a propyl or a methyl group has been found to be deleterious for the antibacterial effect against *S. aureus* and *B. subtilis* [71]. This negative antimicrobial effect was also observed against *L. monocytogenes,* when substituting the phenyl with the propyl moiety [71].

The naphthoquinone 5,8-dihydroxy-1,4-naphthoquinone without chlorine was very active against three Gram-positive (*S. aureus*, *B. subtilis*, and *L. monocytogenes*) and three Gram-negative (*E. coli*, *P. aeruginosa*, and *S. Enteritidis*) strains but in a lower extent against *P. aeruginosa*. The compound 2,3-dichloro-5,8-dihydroxy-1,4-naphthoquinone with chlorine was significantly less active against *E. coli* and *P. aeruginosa* [70]. The presence of gallic or galloyl moieties was found to promote the antibacterial activity of epigallocatechin gallate by inducing damage to the bacterial membrane [72]. The antibacterial action of caffeic acid and their alkyl esters against specific strains of *S. aureus* and *E. coli* showed that longer alkyl side chains were more effective against the Gram-positive bacterium, while caffeic acid esters with medium length alkyl side chain were more effective against the Gram-negative bacterium which was also far less susceptible to caffeic acid and its esters [73].

## 4. Enhancement of Antimicrobial Activity of Plant Derived Polyphenols by Biochemical Methods

Higher potency of the antimicrobial or their use in low concentration is preferred in food application to avoid any changes in the organoleptic properties and minimize interaction with the complex food matrices [14,74]. The following biochemical means may be used to obtain high potent antimicrobials.

### 4.1. Enhancement Using Ascorbic Acid and Transition Metals

One of the promising approaches to enhance antimicrobial activity by non-enzymatic means can be mild oxidation of plant phytochemicals, particularly polyphenols by reactive oxygen species (ROS), using ascorbic acid (AA) and transition metals (Cu (II), Fe (II), Fe (III) systems. However, the reaction mechanism of AA oxidation in the presence of transition metals is still unclear, and different mechanisms have been proposed.

In one study, the AA and $Cu^{2+}$ reaction were projected to yield one mole of hydrogen peroxide ($H_2O_2$) and one mole of dehydro-ascorbic acid in the pH range of 2.6–9.3 [75]. Additionally, it has been assumed that $H_2O_2$ is formed when AA reacts with Cu(II) in the presence of $O_2$ within a pH range of 2.5–4.0 [76]. At physiological pH (7.0), $H_2O_2$ production increased when the copper ($Cu^{2+}$) concentration was deficient, compared with

AA [77]. It was further postulated that the ascorbate mono-anion would dominate when the pH is over 4.25. This form can further be oxidized with the concomitant reduction of copper II to copper I [78]. At pH higher than 4.25, rapid redox cycling of copper generates superoxide, peroxide, and hydroxyl radicals via a copper assisted Fenton reaction, and at a pH lower than 4.25, the level of superoxide in the solution decreases as superoxide anion reacts with hydrogen to form the hydroperoxyl radical ($HO_2^\bullet$) [78].

The reaction product of AA and $Cu^{2+}$ leads to cleavage of viral and plasmid DNA, which could be withdrawn in the presence of metal chelators such as EDTA, stating that copper plays an essential role in the oxidation of AA [79]. The reaction was also withdrawn in the presence of the catalase enzyme, confirming the fact that $H_2O_2$ is mainly produced in the AA/$Cu^{2+}$ reaction [79]. The hydroxyl radical generated by the AA/$Cu^{2+}$ system is lesser than that generated by the ascorbate/$H_2O_2$ system [80]. It has also been reported that transition metal ions, such as Fe (III) and Cu (II), are reduced by ascorbate. Their lower oxidation states (e.g., Fe (II) and Cu (I), respectively) (Equations (1) and (2) below) may further give rise to Fenton reactions with $H_2O_2$, producing hydroxyl radicals [81].

$$Cu^+ + H_2O_2 \rightarrow Cu^{2+} + OH^\bullet + OH^- \tag{1}$$

$$Fe^{2+} + H_2O_2 \rightarrow Fe^{3+} + OH^\bullet + OH^- \tag{2}$$

In the presence of limited oxygen, polyphenols can be oxidized non-enzymatically [82]. The polyphenols containing a catechol ring are oxidized to semiquinone and benzoquinone radicals (Figure 2). At the same time, the oxygen is reduced to $H_2O_2$, and the whole process is mediated by the redox cycle of $Fe^{3+}/Fe^{2+}$ and $Cu^{2+}/Cu^+$ [83]. Recent reviews have illustrated the antimicrobial action of quinone and its derivatives [84–86]. It may be suggested that, in presence of ascorbic acid and transition metals, redox oxygen species may be generated, which may oxidize the polyphenols to corresponding quinones or benzoquinones, enhancing antimicrobial activity [87,88].

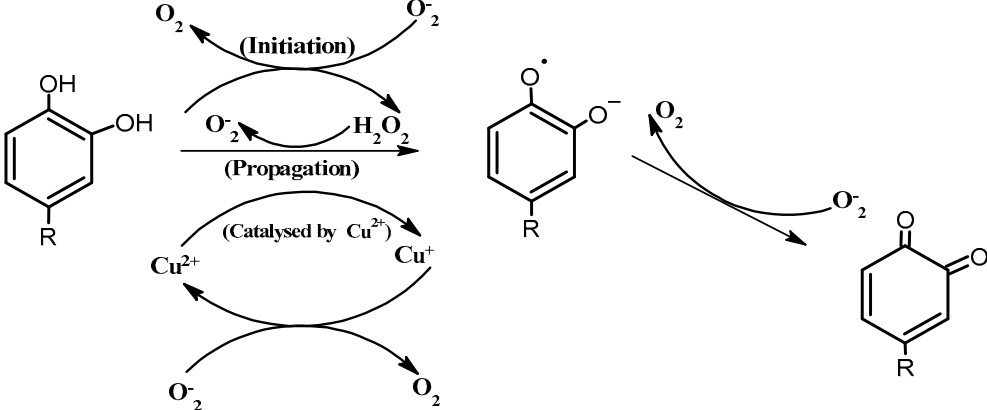

**Figure 2.** The figure illustrates the oxidation of the catechol ring containing polyphenols, which is oxidized to benzoquinone radical with an intermediate formation of semiquinone in presence of $Cu^{2+}$ catalyst. The reaction starts with the initiation step of $O_2$ being converted to $O_2^-$, followed by a propagation step during which $O_2^-$ is converted to $H_2O_2$ [89].

Phenolics from plants have been combined with other substances, sometimes referred to as adjuncts, such as transition metal ions or vitamin C to enhance phenolic efficacy [90,91]. Enhancement of antimicrobial activity was observed in the case of pomegranate rind Ext by Cu (II) alone or with both Cu (II) and AA combinations against many bacterial strains [92,93]. The addition of copper (II) sulfate and AA ascertained the enhancement of antimicrobial activities of whole and sub-fractionated white tea against *S. aureus* [94]. Enhanced antimicrobial activity against *S. aureus* and *E. coli* was also observed by the addition of the AA to the (+)-catechin–copper (II) mixture [95]. A recent study revealed

the antibacterial effect of AA against *S. enterica* subsp. *Enterica* serovar Typhi and *Vibrio fluvialis* could be enhanced when applied in a combination with linalool and copper [96]. However, the mechanism behind the enhancement in antimicrobial activity due to the addition of transition metals and AA to the plant polyphenols has not been appropriately elucidated yet.

### 4.2. Enhancement Using Degradation Products of Ascorbic Acid in an Ethanolic Solution

In aqueous (AQ) systems, AA is very unstable and efficiently degraded both aerobic (AB) and anaerobically. The degradation process of AA is complex and involves many oxidation–reductions and intermolecular rearrangement reactions. The degradation of AA via AB and anaerobic (AAB) pathways depends upon oxygen, heat, light, storage temperature, and time [97,98]. However, degradation of AA mainly results in the formation of volatile and brown products via self-degradation and non-enzymatic browning. The most commonly reported terminal products resulting from the AB degradation of AA and dehydroascorbic acid in acidic AQ conditions (pH 1–3), were found to be 3-hydroxy-2-pyrone and 2-furoic acid [99]. These are also amongst the highest yielding products depending on the conditions utilized. Heat-induced (60–100 °C) AB degradation of different solutions of AA and dehydroascorbic acid demonstrated that both 3-hydroxy-2-pyrone and 2-furoic acid were the main degradation products of AA (Figure 3A). This included decarboxylation of 2,3-diketogulonic acid with the formation of xylosone, a mechanism already reported by the authors of [100], followed by a multi-step conversion of xylosone to the terminal products via oxidation, dehydration, and/or ketoenol tautomerism [100]. Xylosone is a gateway to numerous degradation products, and its presence as an AA or dehydroascorbic acid degradation product has been confirmed by several studies [101,102].

**Figure 3.** Mechanism of aerobic and anaerobic degradation of ascorbic acid. (**a**) aerobic degradation of ascorbic acid with the primary terminal product of degradation as 3-hydroxy-2-pyrone; (**b**) anaerobic degradation of ascorbic acid with the primary terminal product as furfural [103].

The term "AAB degradation" of AA refers to ascorbic acid degradation to some terminal product via a mechanistic pathway that does not require an oxidation step. The AAB degradation of AA does not generally require the removal of $O_2$ from a reaction system; however, the lowering of oxygen concentration has the advantage of limiting the competing AB reactions [103]. The degradation of AA by reaction with oxygen occurs faster than its AAB degradation [104]. However, the AAB degradation rate can increase considerably with higher temperatures [104]. The pH level is also known to influence the rate of AAB degradation of AA, which increases as the pH is raised from 2.3 to 4.0 [105].

Regardless, the most commonly reported terminal product for the AAB degradation of AA is furfural [99,106]. The general mechanism to describe the formation of furfural is shown in (Figure 3B) and involves initial ring cleavage and hydration of AA rather than oxidation [107]. The subsequent steps require decarboxylation, acid-catalyzed dehydration, and cyclization. Several studies have shown that the formation of furfural is favored at lower pH values [108,109].

Another mechanism has been reported for the acid-catalyzed AAB degradation of AA in methanol, forming a bicyclic structure similar to dehydroascorbic acid. It then undergoes dehydration and decarboxylation via dihydrofuran-type intermediates to afford furfural [110]. The authors proposed that the mechanism would be equally valid in AQ systems but did not provide evidence.

Not all scholars agree that furfural is exclusive to the AAB degradation of AA. Instead, it has been suggested that it can be formed via an oxidatively generated dehydroascorbic acid degradation pathway [111,112]. Another product unique to the AAB pathway is, 4,5-dihydroxy-2-ketopentanal (otherwise known as 3-deoxypentosulose) generated upon storage of AA at pH 3.5, although the storage temperature was not so relevant to wine conditions (120 °C for 2 h) [113]. This compound has been proposed as an intermediate in the formation of furfural during the AAB degradation of AA. The intermediate furan compounds which were generated during AA degradation have been found to inhibit the proliferation of *S. typhi* and *B. subtilis* to different extents [114]. The MIC values of furfural and furoic acid (terminal products of AA degradation) against *B. subtilis* and *S. typhi* were 0.027, 0.015, and 0.029, 0.009 μM, respectively [114]. Recent studies have reported antimicrobial action of furoic acid and furan compounds [115,116] The reported end product of AB and AAB degradation might have potency in enhancing the antimicrobial activity of plant Ext. However, to date, there is no evidence of the enhancement of the antimicrobial activity of plant Ext using the degradation product of AA.

### 4.3. Enhancement Using Maillard Reaction Products

The Maillard reaction (MR) is a heat-induced browning reaction widely employed in various fields in the food industry that has been used as an effective method for protein modification and the production of remarkable changes in the structure and bioactivity of proteins [117,118]. In particular, MR products (MRPs) have been shown to have significant antibacterial activities against a wide range of bacteria, with lower toxicity than antibiotics [119,120]. The MRPs possess many intermediate products (Figure 4), such as aldehydes, ketones, and heterocyclic compounds, which can effectively inhibit the growth of some Gram-positive and Gram-negative bacteria [119].

MR products may have the potential to exhibit a synergistic antimicrobial effect in conjunction with phytochemicals from plant compounds, thus enhancing their potency ( i.e., lowering the MIC value of the plant-derived compounds) [121]. The plant-derived polyphenols have been found to have additive, synergistic antimicrobial effects with the intermediate products of the MR such as diacetyls, carbonyls, and furfural [122,123]. Methylglyoxal (one of the intermediate products of MR) and catechin have been reported to positively affect antibacterial activity [124]. However, the antimicrobial activity of plant Ext or plant phytochemicals using MRPs is not yet reported, and further research is needed in this area.

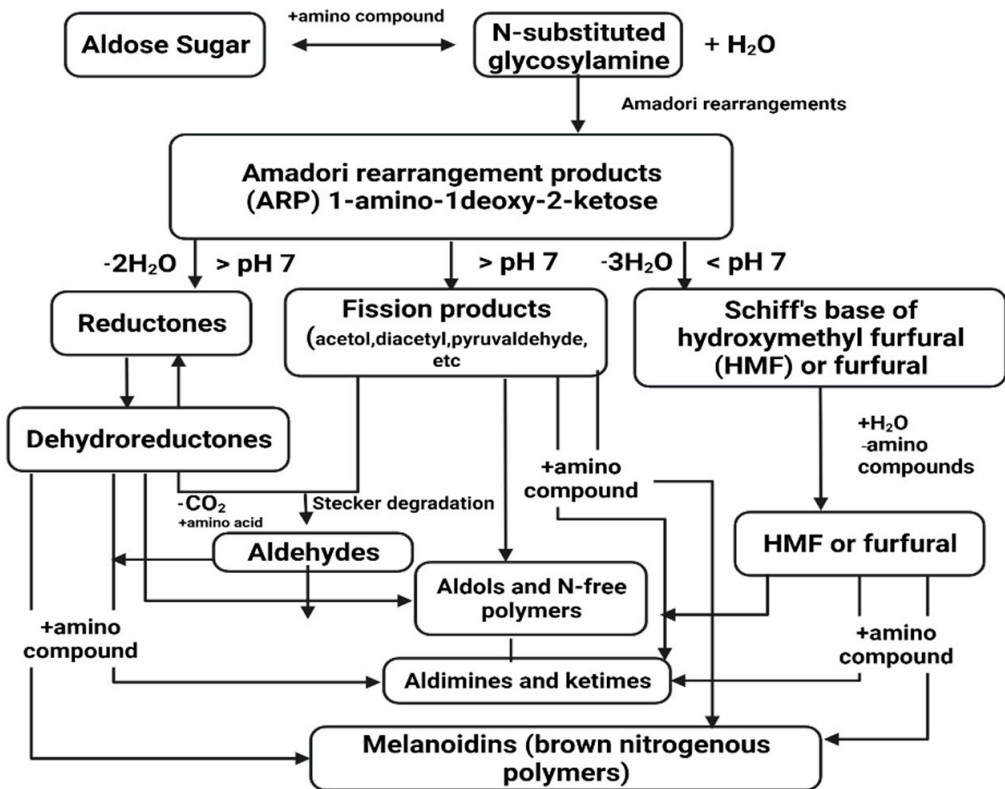

**Figure 4.** Scheme of Maillard reaction in different pH levels consisting of early stage with the formation of the ARP products and Schiff base, followed by the advanced stage, consisting of fission reaction, Stecker degradation, resulting in formation of advanced glycated products with a final stage of oxidation, condensation, cyclization, and rearrangement resulting in melanoidin polymers.

*4.4. Enhancement Using Laccase–Mediator System*

The oxidation of organic compounds to produce functionalized molecules is essential in organic synthesis [125]. Controlled enzymatic oxidation or hypo-oxidation can yet be another approach toward enhancing the antimicrobial activities of phytocompounds. Increased bioactivities have been observed by the biochemical transformation of triterpenes using oxidative enzymes [126]. Oxidoreductive enzymes such as laccase and peroxidise can transform phenols through oxidative coupling reactions with the production of polymeric products by self-coupling or cross-coupling with other molecules.

Laccase (EC 1.10.3.2), is a multi-copper oxidase that couples the four-electron reduction of oxygen with the oxidation of a broad range of organic substrates, including phenols, methoxy-substituted phenols, anilines, aryl diamines, hydroxyindoles, benzenethiols and inorganic/organic metal compounds by a one-electron transfer mechanism, making this green enzyme useful for carrying out several types of oxidative reactions [127–132]. Laccases use $O_2$ as the electron acceptor to remove protons from the phenolic hydroxyl groups. This reaction gives rise to radicals that can spontaneously rearrange, which can either lead to the fission of C–C or C–O bonds of the alkyl side chains or the cleavage of aromatic rings [128]. The oxidation of a reducing substrate by laccase involves losing an electron and forming a free radical [129]. This radical is, in general, unstable and may undergo further laccase-catalyzed oxidation (e.g., quinone from phenol) or non-enzymatic reactions (e.g., hydration, disproportion, or polymerization) [132]. The electron transfer from the substrate to copper is controlled by the redox potential difference. The rate of substrate oxidation by laccase, which has high redox potential, is higher if it has a lower redox potential.

Enzymatic polymerization of phenolic compounds (catechol, resorcinol, and hydroquinone) has been carried out using laccase [133–135]. Intermediates (quinones) formation in the first stage of oxidation with further oxidation reaction, forming colored polymers,

was observed while evaluating the polymerization and the structures of the polymers by UV–Vis and Fourier transform infrared spectroscopy [134]. Changes in the color of flavonoids due to oxidation by the laccase enzyme were due to the polymerization and linkage of the quinones (Figure 5) formed as an intermediate [133]. Laccase oxidation of caffeic acid and isoeugenol was shown to enhance their antimicrobial activity against *S. aureus* and *E. coli* in liquid media [135]. Some low molecular weight phenolic compounds are usually produced as a result of oxidative metabolism by C ring cleavage of catechin and epicatechins [136]. The antimicrobial properties of one of these low molecular weight polyphenols, 3,4 dihydroxy benzoic acid have been well established [137,138]. Moreover, dimers and polymers of flavonoids have also been found to have superior antimicrobial effects in comparison to the parent monomer [139–141]. It has been suggested that the toxicity of the laccase-treated olive Ext can be due to the presence of phenolic compounds such as ortho-benzoquinones, quinonoid, or oxidative coupling polymers, which results because of Lac treatment is more toxic than the parent compounds [142]. Many studies have suggested polymerization of the phenolics using the Lac enzyme [143,144]. Enhanced antimicrobial activity of the resulting oligomers and polymers has also been reported [145,146].

**Figure 5.** Mechanism of catechin oxidation by laccase where catechin is polymerized into polycatechin with the intermediate formation of semiquinone radicals and orthoquinone.

Although laccase can oxidize a wide range of substrates, some substrates of interest cannot be oxidized directly by laccases, either because they are too large to fit into the enzyme active site or because they have an exceptionally high redox potential. By mimicking nature, it is possible to overcome this limitation with the addition of so-called "chemical mediators", which are suitable compounds that act as intermediate substrates for the laccase, whose oxidized radical forms can interact with the bulky or high redox-potential substrate targets [131]. Laccase–mediator system (Figure 6) has found its immense application in the degradation of lignin. Small redox molecules such as 3-hydroxyanthranilic acid (HAA) might act as "electron shuttles" between the enzyme and lignin and cause polymer de-branching and degradation [131]. Some examples of laccase mediators extensively used are 2,2′-azino-bis-(3-ethylbenzothiazoline-6-sulphonic acid) (ABTS), N-hydroxybenzotriazole (HBT), N-hydroxyphtalimide (HPI), violuric acid (VLA), N-hydroxyacetanilide (NHA), methyl ester of 4-hydroxy-3,5-dimethoxy-benzoic acid (syringic acid), and 2,2,6,6-tetramethyl piperidine-1-yloxy (TEMPO) [131].

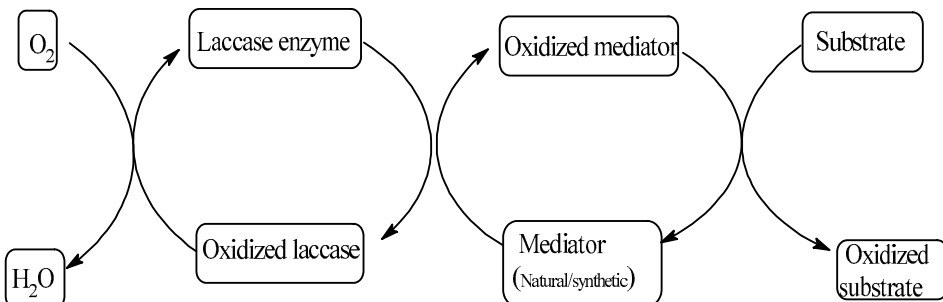

**Figure 6.** Schematic representation of laccase-catalyzed redox cycles according to which laccase is oxydize in presence of oxygen, itself oxidizing the mediator, and the oxidized mediator further initiates the oxidation of the substrate, with high redox potential [131].

Laccase is capable of oxidizing unreactive iodide to reactive iodine [147]. Phenolic compounds such as vanillin, which resembles substructures of softwood lignin, can be directly iodinated by reacting with laccase and iodide, resulting in compounds with antifungal activity [147]. The addition of redox mediators "acetosyringone" in catalytic concentrations increased the rate of iodide oxidation by ten-fold and the yield of iodo-vanillin by 50% [147].

Functionalization is the process of adding new functions, features, capabilities, or properties to material by changing the surface chemistry of the material [148]. In one study, functionalization of chitosan with phenolic acids such as caffeic acid or gallic acid using laccase from *Trametes versicolor* formed a product with enhanced antimicrobial activity against *E. coli* and *L. monocytogenes* [149]. The authors proposed the functionalization of chitosan with the phenolic acids by laccase catalyzed oxidation of phenolic acids to electrophilic o-quinones, which undergo a new oligomer/polymer-forming structure originated by C–C coupling between the benzene rings and C–O–C coupling involved with the phenolic side chains.

### 4.5. Enhancement Using Peroxidase Enzyme

The peroxidases (EC 1.11.1.7) are heme proteins and contain iron (III) protoporphyrin IX (ferriprotoporphyrin IX) as the prosthetic group. They come under the class of oxidoreductases that catalyze the oxidation of a wide range of molecules, using peroxide as an electron acceptor [150]. The reduction of peroxides at the expense of electron-donating substrates makes peroxidases useful in several industrial and analytical applications. The common overall reaction of the peroxidases can be written as in the following Equation (3), where RH is a suitable peroxidase substrate and R is a free-radical product derived from it as follows:

$$2RH + H_2O_2 \rightarrow 2R^{\bullet} + 2H_2O \tag{3}$$

Oxidation reactions carried out by peroxidase may be one-electron or two-electron oxidation. A classic example of one-electron oxidation is guaiacol assay through which guaiacol is oxidized to a free radical that undergoes a subsequent radical–radical combination to give a colored dimeric product. The dimerization can occur between two ring carbon atoms or by adding the oxygen of one phenoxy radical to the ring carbon of the other [151]. Two-electron oxidation is rare for most peroxidase enzymes. The example under this type is the oxidation of halide and pseudo-halide ions, specifically $I^-$, $Br^-$, $Cl^-$, and $NCS^-$. The oxidation of $I^-$ and $NCS^-$ is common for the peroxidases [152]. Many reactions are known in which oxidation of a substrate by peroxidases produces a two-electron oxidized product. These reactions can be rationalized by forming a free radical metabolite, followed by the second oxidation of the free radical to the final observed product.

Lactoperoxidase, together with thiocyanate ions and hydrogen peroxide generates, hypothiocyanite ions and the oxidized product which is known as the lactoperoxidase system. The oxidized effect possesses a broad spectrum of antimicrobial activity. Hence,

much attention has been paid to the lactoperoxidase system [153], e.g., lactoperoxidase systems mediated by oxidized β-carotene/SCN⁻ cycling lead to enhanced antimicrobial effects [154].

*4.6. Future Perspective*

Structural changes in polyphenols by biochemical modifications can be elucidated through elemental analysis and spectral data (IR, $^1$H NMR). Furthermore, crystal, and molecular structures of the potential metal flavonoid or phenolic acid complexes can be identified by using single-crystal X-ray diffraction data. By electron paramagnetic resonance spectroscopy, transient oxidation species could also be identified. Morphological study of the bacterial and fungal cells triggered by biochemically modified polyphenols can reveal the possible antimicrobial effect on the cells. Membrane potential study may indicate the mechanism by which the antimicrobials could have affected the membrane permeability. Ultimately, antibacterial effects against food pathogens can be carried out to understand the scope of biochemically formulated antimicrobials in food applications.

## 5. Conclusions

Polyphenols are widely and easily available bioactive compounds that have health benefits. Despite their benefits, their use in food applications is limited, due to their low potency and high concentration needs. This review showed different biochemical means of improving the antimicrobial property of plant-based polyphenols. Considering the structural importance of the antimicrobial properties of polyphenols, this report also showed the significance of different classes of polyphenols as active antimicrobial agents. Although the mechanisms of the biochemical methods involved in enhancing the antimicrobial activity of plant polyphenols have been explained in this review, the possible structural and functional modifications that these biochemical methods may bring about in the modified polyphenols have not yet been established completely. Therefore, further validation of these methods through high-throughput techniques is crucial.

**Author Contributions:** Conceptualization: L.P., A.D.-S.; writing—original draft preparation: L.P., A.D.-S.; writing—review and editing: L.P., A.D.-S.; visualization: A.D.-S.; supervision: A.D.-S; All authors have read and agreed to the published version of the manuscript.

**Funding:** This research received no external funding.

**Institutional Review Board Statement:** Not applicable.

**Informed Consent Statement:** Not applicable.

**Data Availability Statement:** Not applicable.

**Conflicts of Interest:** The authors declare no conflict of interest.

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
