# Peer review of "Recent Advancements in Enhancing Antimicrobial Activity of Plant-Derived Polyphenols by Biochemical Means"

_horticulturae, doi:10.3390/horticulturae8050401_

Round 1

Reviewer 1 Report

Review report for the article

Horticulturae-1681162

In Abstract, Line no 21,  Please state the conclusive outcome of this study in abstract?

Line no 89. Unbold the phenolic acids.

Line no 92, Unbold the word hydroxycinnamic acids.

Line no 93. Write full form of B. cereus.

Line number 95, Bacillus cereus should be B. cereus. As try to write first time full form and later on abbreviated form.

Line number 99, write full form of A. flavus and A. parasiticus        .

Line no 99, Unbold the word furocoumarins.

Line number 101, Please write full for L. monocytogenes,.

Line number 104, Rewrite this sentence.

Line number 108, What does your mean about Ext.

Line number 111, Remove the full stop.

Line number 125, Sentence should be closed with full stop.

Line number 128, Remove the full stop before the reference 37.

In Line number 142, What do you mean for THIPMC.

In Line number 144, What do you mean for MRSA.

Line number 159, Sentence should be like “tetraflavonoids without OH on the C ring showed moderate activity  against E. coli”.

Line number 204. Remove the full stop.

Line number 210. Streptococcus sobrinus should be S. sobrinus.

Line number 234. Staphylococcus aureus, Bacillus subtilis, and Listeria monocytogenes should be write first time fully after then it should be abbreviated.

Line number 250. Full form AA.

Line number 277. What do you mean for Equation 1 and 2. If described then where???

Line number 277. Unblod the Figure 2 and others as find in bold.

Line number 324. Sentence close by full stop.

Please provide high quality of Figure 4.

Line no 388. Please unbold the “Laccase”.

Line no 458. Please unbold “The peroxidases”.

Line no 471. Please unbold this “dimeric product” and “The dimerization”.

Line no 477. Please unbold this “free radical metabolite”.

References should be according to the journals guideline. And follow same format for all the references.

English language should be improved in throughout the manuscript.

Author Response

Dear reviewer,

Thank you very much for taking the time to read this manuscript. We have read and corrected each of the points you have described in your critique. You will find an attached file detailing each of the observations you have made.

Kind regards

Reviewer 2 Report

Article entitled “Advances in enhancing antimicrobial activity of plant-derived polyphenols by biochemical means” needs some improvements in the contents and structure of the manuscripts

  1. The abstract section's last para needs to rewrite as it is not clear: This review is intended to illustrate the importance of polyphenols as antimicrobials and to portray the mechanisms behind some specific biochemical systems. Ultimately, the objective of this review is to make evident efficient and environmentally friendly methods in the development of antimicrobial agents with potential uses in the food industry.
  2. Most of the reference cited in the articles is too old. It is advised to cite the most recent development in the area of antimicrobial activity of plant-derived polyphenols by biochemical means. The clear indication also needs to discuss about different methods of extraction of polyphenols from plants and also discuss the enhancement procedure.
  3. The author can modify the title of the manuscript by incorporating 10-20 references from the past 3 years like “Recent Advancements in enhancing the antimicrobial activity of plant-derived polyphenols by biochemical means”.
  4. The author can give a more clear view or graphical abstract on different mechanisms of actions of antimicrobial activity of plant-derived polyphenols.

Author Response

Dear reviewer,

Thank you very much for taking the time to read this manuscript. Below you will find the answers we have given to your comments.

  1. The abstract section's last para needs to rewrite as it is not clear: This review is intended to illustrate the importance of polyphenols as antimicrobials and to portray the mechanisms behind some specific biochemical systems. Ultimately, the objective of this review is to make evident efficient and environmentally friendly methods in the development of antimicrobial agents with potential uses in the food industry.

The last paragraph is modified and changed to “This review has portrayed the importance of plant polyphenols, their role as antimicrobial, mechanism of the biochemical methods and the ways these methods may be used in enhancing the antimicrobial potency of the plant polyphenols. Ultimately, this study may act as a base for the development of potent antimicrobial agents which may find its use in food application”

  1. Most of the reference cited in the articles is too old. It is advised to cite the most recent development in the area of antimicrobial activity of plant-derived polyphenols by biochemical means. The clear indication also needs to discuss about different methods of extraction of polyphenols from plants and also discuss the enhancement procedure.

The older citation in the “in antimicrobial activity of plant-derived polyphenols by biochemical means” are replaced and method of extraction of polyphenol is also mentioned in the “plant polyphenols as antimicrobial” section. The removed citations are indicated in the comment section.

  1. The author can modify the title of the manuscript by incorporating 10-20 references from the past 3 years like “Recent Advancements in enhancing the antimicrobial activity of plant-derived polyphenols by biochemical means”.

New references ae incorporated from past 3 years and the title is modified into Recent Advancements in enhancing the antimicrobial activity of plant-derived polyphenols by biochemical means”. Recent references are also added all through the manuscript.

  1. The author can give a clearer view or graphical abstract on different mechanisms of actions of antimicrobial activity of plant-derived polyphenols.

Graphical abstract on different mechanisms of actions of antimicrobial activity of plant-derived polyphenols is prepared and provided in the form of  “graphical abstract.png” file.

Kind regards

Round 2

Reviewer 2 Report

Revised version of manuscript is fine and significantly improved. I appreciate authors for the the revised version of manuscript so it can be taken into consideration for possible publication.